# Contribution of Agro-Physiological and Morpho-Anatomical Traits to Grain Yield of Wheat Genotypes under Post-Anthesis Stress Induced by Defoliation

**Vesna Kandić** [1], **Jasna Savić** [2], **Dragana Rančić** [2,*] and **Dejan Dodig** [1]

[1] Maize Research Institute Zemun Polje, 11080 Belgrade, Serbia; vkandic@mrizp.rs (V.K.)
[2] Faculty of Agriculture, University of Belgrade, 11080 Belgrade, Serbia; jaca@agrif.bg.ac.rs
* Correspondence: rancicd@agrif.bg.ac.rs

**Abstract:** Post-anthesis drought affects wheat production worldwide, primarily through the limitation of grain filling. The enhanced remobilization of stem reserves induced by drought can provide considerable carbon sources for grain filling. The aim of this study, which was part of an ongoing wheat-breeding program targeting drought tolerance, was to assess the ability of 20 contrasting common wheat genotypes (2 cultivars, 8 F6:7 families (FAM), and 10 parent genotypes (PAR) used to make the families) to remobilize stem dry matter under water-stressed conditions simulated via defoliation 10 days after anthesis, and to estimate the contribution of stem dry matter remobilization to grain weight. In two-year field trials, the genotypes were scored for agro-physiological and peduncle morpho-anatomical traits. Stem reserve contribution to grain weight per spike was significantly enhanced in defoliated plants but did not differ amongst the groups of genotypes. F6:7 families had higher grain-filling rate and 1000-grain weight but without improvement in grain weight per spike under defoliation compared with parental groups. The total area of chlorenchyma, phloem-area-related traits, and stem reserve contribution to grain weight were positively associated with grain weight per spike under defoliation, whilst in both treatments, the grain-filling rate was determined by stem height. These results imply that the grain-filling rate is a trait desirable for drought tolerance that can be improved during the breeding process.

**Keywords:** drought; wheat; stem; peduncle; dry matter remobilization

## 1. Introduction

Wheat (*Triticum aestivum* L.) is a staple food and one of the four major crops grown worldwide, providing around 20% of calories and protein for the global population [1]. As one of the most important environmental stresses, drought reduces wheat grain yield in many parts of the world. Recent analyses by Helman and Bonfil [2] indicated that wheat production is adversely affected by warming and drought in the world's leading wheat-producing countries. Moreover, it is estimated that global wheat production will decrease by 6% for each °C of further temperature increase and becomes more variable over space and time [3], due to potential rainfall variability with frequent droughts. Under future climate scenarios, there is a tendency that total precipitation might decrease in the Mediterranean region between 10% and 40% [4], and a similar range of future decline in precipitation is projected for the Balkan region [5,6]. The wheat production area in Serbia is around 600,000 ha, and over the past ten years, the average grain yield has varied from 4 t/ha up to 5.7 t/ha [1] and is often strongly affected by lack of precipitation during the sowing and grain filling periods. Therefore, the development of improved wheat genotypes tolerant to drought for sustainable wheat production has become more challenging than ever.

The terminal drought that occurs after anthesis, commonly in combination with heat stress, adversely affects wheat grain set and the duration of grain filling [7], thus causing





wheat grain weight and yield reduction [8,9]. In wheat, grain filling depends on three main sources: the current assimilates produced by photosynthesis in leaves and stems, the mobilization of the stored carbohydrates and nitrogen-containing compounds within these organs and their subsequent transport to the ear and grains, and the assimilates produced by the ear [10]. Under optimal conditions, the major source of assimilates is considered to be the flag leaf [11]. Post-anthesis drought stress decreases photosynthetic carbon assimilation but increases the remobilization of non-structural carbohydrates from the vegetative tissues to the grain [12,13]. Under such conditions, the carbohydrates stored in the stem become the predominant material transported to grains [14,15].

Under favorable conditions, reserve mobilization could potentially contribute to around 20% of grain dry weight [16], though the proportion increases considerably under drought conditions [17]. For example, Zhang et al. [18] demonstrated that, under water-stressed conditions, the contribution of the ear and peduncle to grain weight was as much as 73%. The development of the vascular system in the peduncle is essential for transporting assimilates to the grain during grain filling [19], and this internode always maintains a higher water potential than the flag leaf under drought stress or high temperatures [20]. Therefore, a better understanding of the relationship between grain filling and both stem morphology and peduncle anatomical traits is crucial for the development of wheat better adapted to post-anthesis water stress.

This study aimed to assess the ability of contrasting wheat genotypes (F6:7 families and their parents) to remobilize stem dry matter under water-stressed conditions simulated via defoliation during grain filling and its contribution to grain weight. We also evaluated the association of peduncles' morphological and anatomical traits with grain weight and other important traits to see whether any progress has been made in a wheat-breeding program targeted toward improved drought tolerance.

## 2. Materials and Methods

### 2.1. Genotypes

This study was conducted as part of the ongoing wheat-breeding program at the Maize Research Institute Zemun Polje (MRIZP). The experimental material comprised three groups of winter or facultative common wheat (*Triticum aestivum* L.) cultivars and breeding lines: standard genotypes (2 entries used in the wheat-breeding program at MRIZP), F6:7 families (8 entries) and parent genotypes used to make the F6:7 families (10 entries). The wheat genotypes evaluated in the present study were selected based on previously conducted two-year trials in which initially sixty-one, and then forty-four wheat genotypes were evaluated [7,21–23]. Genotype names, origin, and parentage are given in Table 1.

### 2.2. Field Trials

2.2.1. Experimental Site

The field trials were carried out at the experimental site of MRIZP (44°52′ N and 20°19′ E, 82 m a.s.l.), Serbia, in two consecutive wheat growing seasons: 2012–2013 and 2013–2014. The climate in the area is moderately continental with cold winters and hot, often dry summers; variation in spring and summer air temperature and precipitation over seasons is high. The soil at the experimental site was chernozem, with a humus content of 3.2% [24]. Data for mean daily temperature, precipitation, and number of days with maximum temperature > 30 °C for two growing seasons are provided in Table 2. These two seasons were characterized by contrasting weather conditions; the accumulated rainfall during grain filling was much higher in the second season.

**Table 1.** Name and origin of 20 wheat genotypes analyzed, with parentage for F6:7 families.

| Entry Name | Origin | Parentage |
|---|---|---|
| Standard genotypes | | |
| Zemunska rosa 1 | Serbia | |
| Zemunska rosa 2 | Serbia | |
| Families | | |
| MRI S4/I | Serbia | ZGKT 159/82 × Donska semidwarf |
| MRI S10/I | Serbia | Highbury × Mexico 3 |
| MRI D3/I | Serbia | Brigand × Pobeda |
| MRI D6/I | Serbia | Bezostaya 1 × Florida |
| MRI D10/I | Serbia | Lambriego Inia × Bezostaya 1 |
| MRI D19/I | Serbia | NS 46/90 × Pobeda |
| MRI D20/IP | Serbia | Lambriego Inia × Florida |
| MRI D22/I | Serbia | Lambriego Inia × NS 46/90 |
| Parents | | |
| Donska semidwarf | Russia | |
| Brigand | Great Britain | |
| Highbury | Great Britain | |
| Florida | USA | |
| NS 46/90 | Serbia | |
| Bezostaya 1 | Russia | |
| Lambriego Inia | Chile | |
| Mexico 3 | Mexico | |
| Pobeda | Serbia | |
| ZGKT 159/82 | Croatia | |

**Table 2.** Weather conditions during two winter wheat growing seasons.

| Months | Growing Seasons | | | | | |
|---|---|---|---|---|---|---|
| | 2012–2013 | | | 2013–2014 | | |
| | Mean Temperature (°C) | Days >30 °C | Precipitation (mm) | Mean Temperature (°C) | Days >30 °C | Precipitation (mm) |
| November | 9.9 | 0 | 22.6 | 9.4 | 0 | 28.7 |
| December | 1.2 | 0 | 43.9 | 2.1 | 0 | 6.7 |
| January | 2.8 | 0 | 54.8 | 4.6 | 0 | 24.6 |
| February | 4.2 | 0 | 50.6 | 7.0 | 0 | 12.9 |
| March | 6.0 | 0 | 87.9 | 9.9 | 0 | 44.5 |
| April | 14.1 | 0 | 27.7 | 13.1 | 0 | 86.9 |
| May | 18.2 | 3 | 98.6 | 16.8 | 0 | 233.4 |
| June | 20.7 | 9 | 39.2 | 20.8 | 6 | 85.6 |
| Mean/sum | 9.6 | 12 | 425.3 | 10.5 | 6 | 523.3 |

2.2.2. Experimental Design and Treatments

Experiments were set up in a completely randomized block design with two replications. Seeds of each genotype were sown in five 1 m long rows, spaced at 20 cm, with a seeding rate of 450 seeds/m$^2$. The experiments were sown in late October in both growing seasons. Standard field management practices were applied until harvest. To prevent a reduction in reserve accumulation and storage capacity in stems, the plots were additionally manually irrigated until the end of stem elongation, when water in the top 0.75 m of soil had decreased to <50% of field capacity. In defoliated plants (DP), all leaf blades were cut off 10 days after anthesis (10 DAA), at the time of the rapid grain filling phase (growth stage

(GS)71–73 [25]). Intact plants were the control treatment (CP). The treatment simulates drought stress by inhibiting current assimilation [26,27].

### 2.2.3. Sampling and Measurements

Plants were tagged for sampling at anthesis as follows: Fifteen uniform plants of each genotype flowering on the same day from interior rows within each replicate were tagged. Five tagged plants were sampled at 10 DAA, and the remaining five control and five defoliated plants were sampled at physiological maturity.

Genotypes were scored for agro-physiological and morpho-anatomical traits. For morpho-anatomical analysis, the main stem samples of 5 plants were taken at 10 DAA. Peduncle (the uppermost internode) length and extruded peduncle (from the flag leaf ligule to the base of the ear), penultimate internode, and total stem lengths were measured. The peduncle's share of the total stem length was calculated as the ratio of its length to the total stem length. Concomitantly, two types of 10 mm long samples were collected from the peduncle: the first from the upper exposed part just below the spike (part of the stem outside of the flag leaf sheath), and the second one from the lower part of the peduncle, just above the node (part of the stem within the flag leaf sheath).

The samples were fixed in 50% ethanol, and transversal sections were obtained manually using razor blades. Microslides were stained with phloroglucinol–HCl [28] and observed under a LEICA 2000 light microscope. The images were captured by a LEICA DC320 camera connected to the microscope, and micrographs were measured in IM1000 software. The following anatomical parameters were measured in both parts of the peduncle: number of small vascular bundles (outer ring of bundles; NSB), number of large vascular bundles (inner ring of bundles; NLB), peduncle diameter (PD), peduncle wall thickness (PWT), the total area of parenchyma per stem transverse section (photosynthetically inactive parenchyma; AP), the total area of chlorenchyma per stem transverse section (AC), the average phloem area in one small vascular bundle (PSB), the average phloem area in one large vascular bundle (PLB), the total area of phloem per stem transverse section (TP), the area of pith cavity (APC) and the area of peduncle wall (AW) (Figure 1). The results obtained for all morpho-anatomical traits are presented as means for the two analyzed parts of the peduncle. Data for traits measured at 10 DAA were pooled from both plants and tagged as control and defoliated.

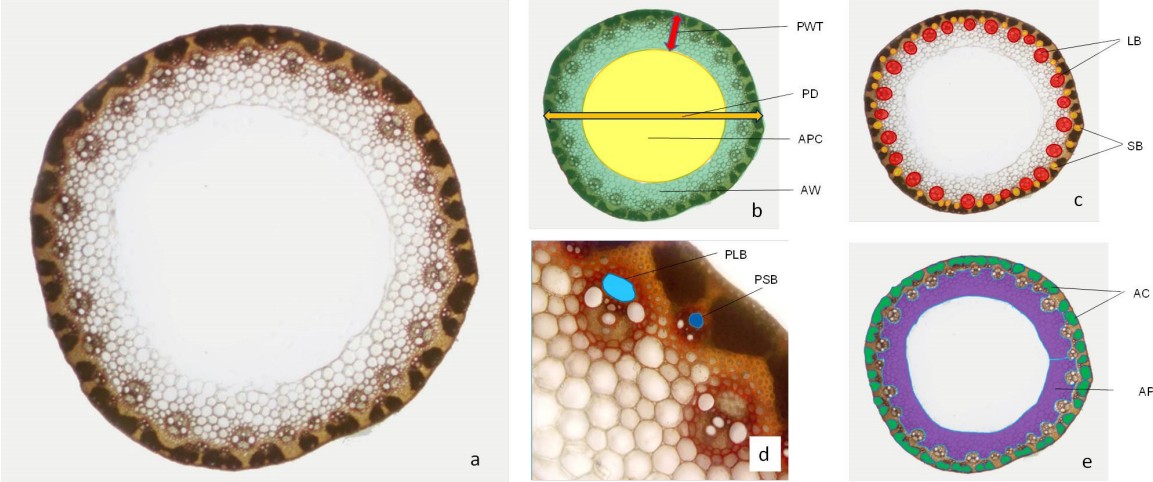

**Figure 1.** Cross-section of wheat peduncle stained with phloroglucinol–hydrochloric acid (**a**); PD—peduncle diameter, PWT—peduncle wall thickness, AW—area of peduncle wall, APC—area of pith cavity (**b**); LB—large vascular bundles, SB—small vascular bundles (**c**); PLB—phloem area in large vascular bundle, PSB—phloem area in small vascular bundle (**d**); AC—area of chlorenchyma on whole peduncle section, AP—area of parenchyma on whole peduncle section (**e**).

Main stem dry biomass (BMS) was measured at 10 DAA and at harvest, after drying at 80 °C for 48 h. The number of spikelets per spike (SPS), grain number per spike (GNS), harvest index (HI; grain weight/total above ground weight), 1000-grain weight (TGW), and grain weight per spike (GWS) were measured at harvest. The amount of mobilized dry matter from stem to grain was estimated as the difference between stem weight at 10 DAA and maturity. Stem reserve contribution to grain yield (SRC) was calculated as its proportional contribution (%) to grain weight. The grain-filling rate (GFR) was estimated as grain weight at maturity divided by duration (expressed as growing degree days), assuming that grain weight at anthesis was zero. Cumulative growing degree-days, from 10 DAA to physiological maturity, were calculated as the sum of mean daily temperatures $((\text{max} + \text{min})/2) - $ base temperature. A base temperature of 0 °C was assumed for the grain filling period [29]. Two stress indices were calculated to characterize the genotype response to stress induced by defoliation. A stress susceptibility index (SSI) was calculated as follows: $\text{SSI} = (1 - (\text{Ts1})/(\text{Tns1}))/\text{SI}$, where Ts and Tns are the grain weights of a genotype under stressed (defoliated) and non-stressed conditions, respectively [30]. The stress intensity index (SI) was estimated from $(1 - (\text{Xs2})/(\text{Xns2}))$, where Xs2 and Xns2 represent the mean grain weight across all the genotypes evaluated under stressed and non-stressed conditions, respectively. A stress tolerance index (STI) was calculated using the following equation: $\text{STI} = (\text{Tns} \times \text{Ts})/(\text{Xns})^2$ [31].

### 2.3. Statistical Analyses

The statistical parameters of mean, maximum, and minimal values and coefficient of variation were used to describe the variability of morpho-anatomical and agro-physiological traits in each treatment. Three-way ANOVA was performed to evaluate the effects of genotype, treatment, and year, and their interaction on agro-physiological traits across two years. One-way ANOVA was performed to study the effect of defoliation on investigated traits for each year, as well as the effect of a group of genotypes on measured traits. The significance of differences between means was tested using Tuckey's test ($p < 0.05$). Principal component analysis (PCA) was used to analyze and visualize the relationships between the observed traits, and a scatterplot was created to visualize wheat genotypes in the coordinate system of the first two PCs [32]. Separate biplots were constructed for the two treatments. A positive correlation between two traits was represented by an acute angle between them, while an obtuse angle represented a negative correlation (cosine $0° = 1$ is the maximum positive correlation, cosine $180° = -1$ is the maximum negative correlation, and cosine $90° = 0$ is an absence of correlation). Statistical analyses were performed using the Minitab 19 software package.

## 3. Results

### 3.1. Stem Height and Peduncle Morpho-Anatomical Traits of Genotype Groups

The three groups STA, FAM, and PAR did not differ significantly in the number of days to anthesis across the two growing seasons (Table 3). For the peduncle's morpho-anatomical traits measured at 10 DAA, the three groups of genotypes did not differ significantly ($p < 0.05$) in peduncle wall thickness, the number of large vascular bundles, phloem area in small vascular bundles, peduncle length, and peduncle extrusion. STA genotypes had significantly lower peduncle diameter, area of parenchyma, area of chlorenchyma, total phloem area (in both small vascular bundles and per peduncle section), and area of peduncle wall compared with both FAM and PAR. In addition, the STA group had a significantly smaller phloem area in large vascular bundles and total phloem area in large vascular bundles, shorter stem and higher peduncle share than FAM, and also a smaller number of small vascular bundles and area of pith cavity in comparison with PAR. The genotypes from FAM and PAR did not differ significantly for any of the measured morpho-anatomical traits. The highest variation amongst the genotypes within FAM and PAR groups was recorded for peduncle share (25.9% and 31.2%, respectively) and the pith cavity area (12.2%) in STA.

**Table 3.** Descriptive statistics for stem height and peduncle morpho-anatomical traits for 20 common wheat genotypes—standards, F6:7 families, and parent genotypes used to make the F6:7 families. Data represent mean values for two growing seasons.

| Trait | Standard Genotypes | | | Families | | | Parents | | |
|---|---|---|---|---|---|---|---|---|---|
| | Mean $\pm$ S.E. | Range | CV (%) | Mean $\pm$ S.E. | Range | CV (%) | Mean $\pm$ S.E. | Range | CV (%) |
| Days to anthesis | 127 [a] $\pm$ 0.0 | 127 | 0.0 | 132.1 [a] $\pm$ 1.16 | 128–137 | 2.5 | 131.7 [a] $\pm$ 1.45 | 127–138 | 3.5 |
| Peduncle diameter (mm) | 2.54 [b] $\pm$ 0.03 | 2.51–2.58 | 1.9 | 2.91 [a] $\pm$ 0.12 | 2.53–3.36 | 11.3 | 2.94 [a] $\pm$ 0.09 | 2.38–3.44 | 9.6 |
| Peduncle wall thickness (mm) | 0.54 [a] $\pm$ 0.14 | 0.53–0.56 | 3.6 | 0.57 [a] $\pm$ 0.02 | 0.51–0.66 | 9.2 | 0.58 [a] $\pm$ 0.02 | 0.49–0.679 | 11.0 |
| Area of parenchyma (mm$^2$) | 2.05 [b] $\pm$ 0.02 | 2.03–2.08 | 1.7 | 2.41 [a] $\pm$ 0.14 | 1.90–2.87 | 15.8 | 2.49 [a] $\pm$ 0.14 | 1.92–3.28 | 18.0 |
| Number of large VBs * | 25.1 [a] $\pm$ 0.2 | 24.8–25.3 | 1.1 | 26.1 [a] $\pm$ 1.07 | 22.1–29.7 | 11.6 | 27.5 [a] $\pm$ 1.3 | 22.9–33.8 | 15.3 |
| Phloem area in large VB ($\mu$m$^2$) | 1500 [b] $\pm$ 49.4 | 1450–1549 | 4.7 | 1807.5 [a] $\pm$ 81.0 | 1562–2177 | 12.7 | 1637.4 [ab] $\pm$ 62.3 | 1378–1988 | 12.0 |
| Area of chlorenchyma (mm$^2$) | 0.24 [b] $\pm$ 0.01 | 0.23–0.25 | 4.3 | 0.34 [a] $\pm$ 0.02 | 0.24–0.44 | 20.3 | 0.35 [a] $\pm$ 0.02 | 0.24–0.46 | 18.0 |
| Number of small VBs | 27.15 [b] $\pm$ 0.3 | 26.9–27.5 | 1.6 | 29.72 [ab] $\pm$ 1.1 | 25.8–33.1 | 10.2 | 34.24 [a] $\pm$ 2.1 | 25.4–45.15 | 19.4 |
| Phloem area in small VB ($\mu$m$^2$) | 161.0 [a] $\pm$ 7.6 | 153.5–168.6 | 6.7 | 184.3 [a] $\pm$ 9.82 | 150–222 | 15.1 | 166.9 [a] $\pm$ 6.1 | 138.2–201.3 | 11.5 |
| Total phloem area in large VBs (mm$^2$) | 0.037 [b] $\pm$ 0.001 | 0.036–0.038 | 5.2 | 0.047 [a] $\pm$ 0.003 | 0.037–0.060 | 19.0 | 0.045 [ab] $\pm$ 0.003 | 0.034–0.060 | 22.5 |
| Total phloem area in small VBs (mm$^2$) | 0.005 [b] $\pm$ 0.000 | 0.005–0.005 | 1.5 | 0.006 [a] $\pm$ 0.0003 | 0.005–0.007 | 15.1 | 0.006 [a] $\pm$ 0.0003 | 0.005–0.008 | 16.5 |
| Total phloem area/peduncle section (mm$^2$) | 0.04 [b] $\pm$ 0.00 | 0.04–0.04 | 4.4 | 0.05 [a] $\pm$ 0.003 | 0.04–0.07 | 18.1 | 0.05 [a] $\pm$ 0.03 | 0.04–0.07 | 20.91 |
| Area of peduncle wall (mm$^2$) | 3.43 [b] $\pm$ 0.013 | 3.42–3.45 | 0.6 | 4.11 [a] $\pm$ 0.194 | 3.29–4.86 | 13.4 | 4.41 [a] $\pm$ 0.21 | 3.17–5.57 | 15.4 |
| Area of pith cavity (mm$^2$) | 1.79 [b] $\pm$ 0.16 | 1.63–1.63 | 12.2 | 2.54 [ab] $\pm$ 0.32 | 1.53–3.75 | 35.5 | 2.70 [a] $\pm$ 0.25 | 1.35–3.76 | 29.4 |
| Stem height (cm) | 88.5 [b] $\pm$ 1.0 | 87.4–89.5 | 1.7 | 96.1 [a] $\pm$ 1.9 | 88.0–103.6 | 5.6 | 88.6 [ab] $\pm$ 3.6 | 72.1–104.4 | 12.9 |
| Peduncle length (cm) | 34.5 [a] $\pm$ 0.2 | 34.3–34.7 | 0.7 | 34.2 [a] $\pm$ 0.9 | 29.5–37.4 | 7.8 | 33.7 [a] $\pm$ 1.4 | 28.3–36.8 | 12.2 |
| Peduncle extrusion (cm) | 15.3 [a] $\pm$ 0.1 | 15.2–15.3 | 0.6 | 12.1 [a] $\pm$ 1.0 | 8.8–16.9 | 25.9 | 12.9 [a] $\pm$ 1.3 | 7.2–19.2 | 31.2 |
| Peduncle share of the total stem length (%) | 40 [a] $\pm$ 0.01 | 39–41 | 3.0 | 37 [b] $\pm$ 0.01 | 32–40 | 7.6 | 39 [ab] $\pm$ 0.01 | 35–40 | 9.6 |

* VB—vascular bundle. Means of the traits for three groups of genotypes followed by the same letter (lower case) are not significantly different ($p < 0.05$).

## 3.2. Differences between Control and Defoliated Plants in Agro-Physiological Traits

Three-way ANOVA revealed the significant main effects of genotype, treatment, and year on all agro-physiological traits, with the exception of the effect of treatment on the number of spikelets per spike and the effect of year on the 1000-grain weight (Table 4). Genotype × year interaction was significant for all the measured traits, while genotype × treatment interaction was not significant for the number of spikelets per spike and harvest index, and treatment × year interaction was not significant for the main stem biomass and the number of spikelets per spike. G × T × Y interaction had a significant effect on all traits except for the number of spikelets per spike.

**Table 4.** Summary of ANOVA of the effect of genotype, treatment, and year on agro-physiological traits for the 20 wheat genotypes (standards, families, and parents).

| Source of Variation | Main Stem Biomass | Number of Spikelets per Spike | Grain Number per Spike | Grain Weight per Spike | 1000-Grain Weight | Harvest Index | Grain-Filling Rate | Stem Reserve Contribution |
|---|---|---|---|---|---|---|---|---|
| | | | | *p* | | | | |
| Genotype (G) | <0.001 | <0.001 | <0.001 | <0.001 | <0.001 | <0.001 | <0.001 | <0.001 |
| Treatment (T) | <0.001 | 0.168 ns | <0.001 | <0.001 | <0.001 | <0.001 | <0.001 | <0.001 |
| Year (Y) | <0.001 | <0.001 | <0.001 | <0.001 | 0.18 ns | <0.001 | <0.001 | <0.001 |
| G × T | <0.01 | 0.12 ns | <0.001 | <0.01 | <0.001 | 0.32 ns | <0.001 | <0.001 |
| T × Y | 0.12 ns | 0.24 ns | <0.001 | <0.001 | <0.001 | <0.001 | <0.001 | <0.001 |
| G × Y | <0.001 | <0.001 | <0.001 | <0.001 | <0.001 | <0.001 | <0.001 | <0.001 |
| G × T × Y | <0.05 | 0.183 ns | <0.001 | <0.001 | <0.001 | <0.001 | <0.001 | <0.001 |

ns—not significant.

In both growing seasons, on average across the three groups, control and defoliated plants differed significantly ($p < 0.05$) in stem dry biomass, grain weight per spike, number of grains per spike, and stem reserve contribution (Table 5). Defoliation decreased stem biomass and grain weight per spike across the two growing seasons by 30.3% and 25.6% in 2012/3 and 2013/4, respectively. Stem reserve contribution to grain weight per spike ranged from 5.5% to 99.0% in control and from 27.8% to 99.0% in defoliated plants and was almost four-fold and two-fold higher in the defoliation treatment in the first and second seasons, respectively. Such high estimated stem reserve contributions to grain weight per spike in some FAM and PAR genotypes were due to their low grain weight per spike. In contrast, the 1000-grain weight and the grain-filling rate were significantly reduced by defoliation in the first growing season only. In both treatments, the harvest index was significantly higher in the first growing season. The majority of traits showed considerably high variation (>20%) in both treatments, with the highest being the stem reserve contribution in the control treatment (around 60%). The differences amongst genotypes in grain number and grain weight per spike, 1000-grain weight, grain-filling rate, and harvest index across the two seasons were pronounced in both treatments (Supplementary Table S1). SSI can be used to compare genetic capacity to sustain yield under stress in the post-anthesis phase, and lower values indicate greater tolerance to drought stress. Genotypic response to tolerate post-anthesis stress differed remarkably between the genotypes. The highest SSI was obtained for genotype MRI D10/1 (0.42), suggesting that this line had a high capacity to maintain yield under stress conditions. SSI ranged from 0.42 to 2.39. The stress tolerance index (STI) can be used to select high-yielding genotypes in both stress and non-stress conditions, and higher values indicate greater tolerance to stress. In our study, STI ranged from 0.27 (MRI D20/IP) to 1.21 (Florida).

**Table 5.** Descriptive statistics for the agro-physiological traits for 20 wheat genotypes in control (CP) and defoliated plants (DP), during two growing seasons, 2012–2013 and 2013–2014.

| Traits | Treatment | Mean ± SE | DP/CP | Range | CV (%) | Mean ± SE | DP/CP | Range | CV (%) |
|---|---|---|---|---|---|---|---|---|---|
| | | | | 2012–2013 | | | | 2013–2014 | |
| Main stem dry biomass (g) | CP | 1.9 aB ± 0.0 | 0.63 | 1.5–2.5 | 14.2 | 2.1 aA ± 0.1 | 0.76 | 1.5–3.0 | 18.5 |
| | DP | 1.2 bB ± 0.0 | | 0.9–2.0 | 20.1 | 1.6 bA ± 0.1 | | 1.3–2.1 | 18.4 |
| Number of spikelets per spike | CP | 21.2 aB ± 0.5 | 0.97 | 16.3–26.7 | 11.1 | 22.6 aA ± 0.5 | 0.95 | 18.8–28.0 | 10.8 |
| | DP | 20.6 aA ± 0.5 | | 16.0–26.7 | 11.3 | 21.4 aA ± 0.6 | | 17.6–28.0 | 11.6 |
| Number of grains per spike | CP | 49.2 aA ± 2.7 | 0.86 | 32.3–73.7 | 24.2 | 49.0 aA ± 3.1 | 0.76 | 35.0–86.2 | 28.5 |
| | DP | 42.5 bA ± 2.2 | | 20.7–62.0 | 23.3 | 37.4 bA ± 3.1 | | 19.4–71.6 | 37.2 |
| Grain weight per spike (g) | CP | 1.8 aA ± 0.1 | 0.72 | 1.1–2.7 | 22.5 | 1.7 aA ± 0.1 | 0.76 | 0.7–2.5 | 27.2 |
| | DP | 1.3 bA ± 0.1 | | 0.6–1.9 | 28.5 | 1.3 bA ± 0.1 | | 0.4–2.1 | 37.1 |
| 1000-grain weight (g) | CP | 38.4 aA ± 1.5 | 0.78 | 24.3–47.9 | 17.3 | 34.5 aB ± 2.1 | 0.95 | 16.4–50.6 | 26.6 |
| | DP | 30.2 bA ± 1.5 | | 20.0–44.7 | 21.9 | 33.0 aA ± 1.6 | | 21.1–52.0 | 21.8 |
| Harvest index (%) | CP | 43.6 aA ± 1.5 | 0.93 | 32.4–62.2 | 15.3 | 37.1 aB ± 1.3 | 0.88 | 18.8–44.0 | 15.7 |
| | DP | 40.5 aA ± 1.8 | | 15.6–51.8 | 20.1 | 32.8 bB ± 2.4 | | 13.1–49.3 | 33.2 |
| Grain-filling rate (mg 100 GDD/day) | CP | 6.2 aA ± 0.3 | 0.77 | 4.0–7.5 | 17.8 | 4.9 aB ± 0.3 | 0.98 | 2.3–7.4 | 25.4 |
| | DP | 4.8 bA ± 0.2 | | 3.4–7.0 | 20.9 | 4.8 aA ± 0.3 | | 3.0–7.5 | 24.4 |
| Stem reserve contribution (%) | CP | 18.8 bB ± 3.5 | 3.76 | 5.2–61.2 | 82.6 | 38.9 bA ± 5.3 | 2.00 | 7.8–99.0 | 61.0 |
| | DP | 70.7 aA ± 4.9 | | 27.8–99.8 | 31.1 | 77.8 aA ± 5.5 | | 34.6–99.0 | 31.8 |

Means of traits in control and defoliated plants within same year followed by the same letter (lower case) are not significantly different ($p < 0.05$). Means of the traits for each year within same treatment followed by the same letter (upper case) are not significantly different ($p < 0.05$).

### 3.3. Differences between Groups of Genotypes in Agro-Physiological Traits

The differences between STA, FAM, and PAR groups of genotypes in agro-physiological traits within each treatment and the differences between treatments for each group of genotypes across the two growing seasons are presented in Figure 2. Variation amongst the SMA, FAM, and PAR groups in agro-physiological traits in the control treatment was relatively small, except for grain number per spike, the 1000-grain weight, and the grain-filling rate. Thus, PAR genotypes had a significantly higher number of grains per spike than STA and FAM, while the 1000-grain weight and the grain-filling rate were significantly higher in FAM than in the PAR group. Under defoliation, PAR genotypes had significantly a higher grain number per spike than FAM genotypes. However, the FAM group had a significantly higher 1000-grain weight and grain-filling rate compared with the PAR group. Similar significant reductions induced by defoliation were recorded for stem biomass (26.3–29.4%) and grain weight per spike (23–27.8%) for all three groups of genotypes. The 1000-grain weight and the grain-filling rate were significantly reduced by defoliation only for PAR genotypes, while grain number per spike was significantly reduced for both FAM and PAR groups. The harvest index and the number of spikelets per spike were significantly lower in defoliated plants only for FAM genotypes.

The stem reserve contribution to grain weight per spike was increased by defoliation in STA and PAR (by 270% relative to the control) and by 230% in FAM.

### 3.4. Relationships among Traits

To summarize the relationships existing between the GWS and other agro-physiological traits and peduncle morpho-anatomical traits, a genotype-by-trait PCA was separately performed for the control and defoliated plants across the twenty genotypes and two growing seasons (Figure 3).

The first two principal components of the PCA explained around 55.7% and 53.1% of the observed variability in control and defoliation treatment, respectively (Figure 3a,b). In the control, grain weight per spike was positively associated with agro-physiological traits such as stem biomass, stem reserve contribution, and harvest index. Similarly, grain weight per spike was closely associated with stem biomass and the 1000-grain weight and grain-filling rate under treatment with defoliation. The association of grain weight per spike with anatomical traits differed between defoliated and control plants. For control plants, the total phloem area in small vascular bundles, phloem area in large vascular bundles, peduncle diameter, and the area of peduncle wall had the strongest influences on grain weight per spike. The grain weight per spike of defoliated plants was associated with the area of chlorenchyma, peduncle diameter, and phloem area in small vascular bundles. The 1000-grain weight appeared to be determined by the grain-filling rate and stem height in both treatments.

The visualization of the 20 wheat genotypes as a scatterplot in the coordinate system of the first 2 PCs for 25 traits is given in Supplementary. A similar score of the genotypes was obtained for both treatments, without a clear distinction between the groups. Two standard genotypes were close to each other in both treatments, and this was the case with four PAR genotypes (Highbury, Mexico 3, ZGKT 159/82, and Bezostaya 1) and four FAM genotypes (MRI S4/I, MRI D22/I, MRI D3/I, and MRI D20/IP).

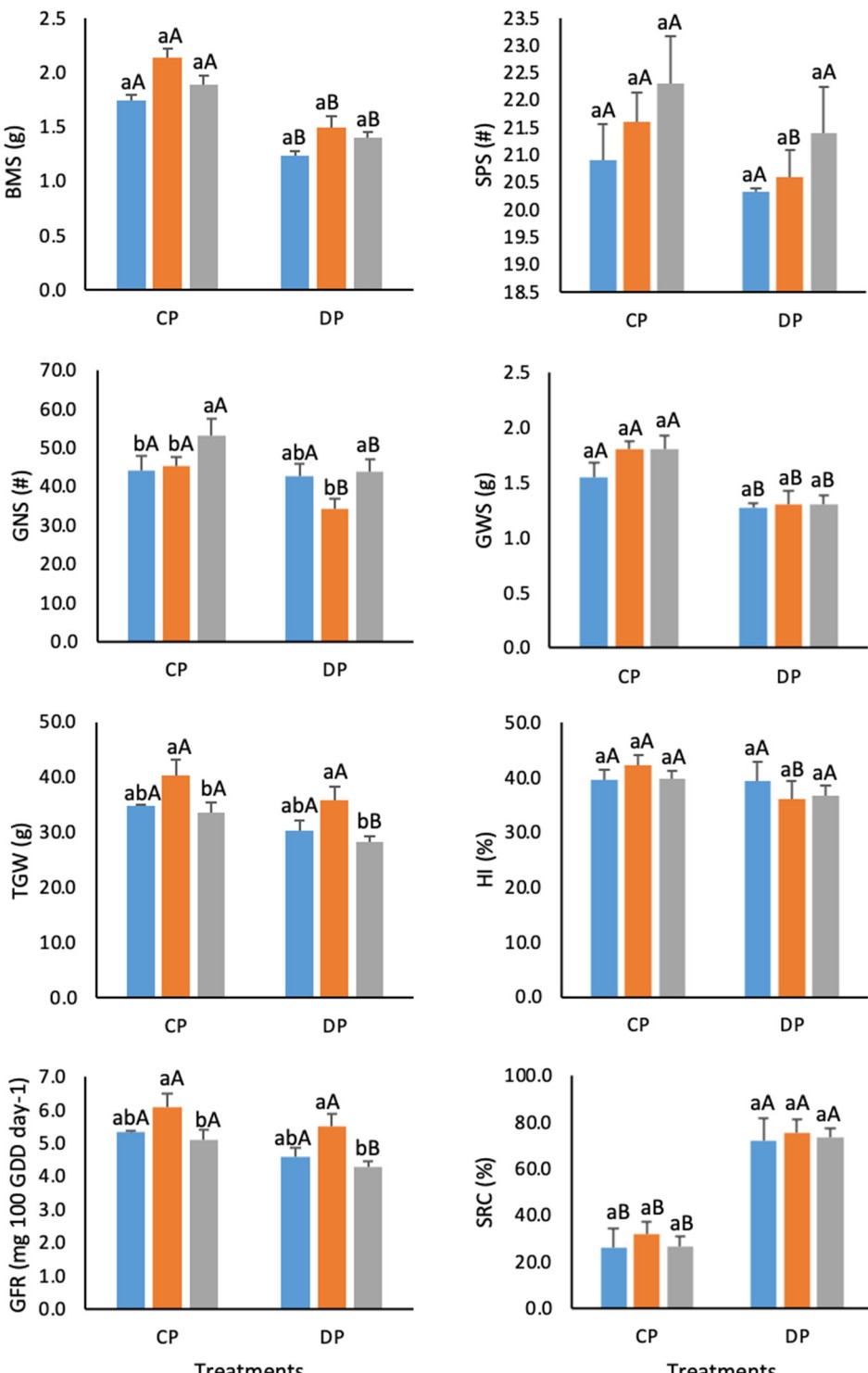

**Figure 2.** Performance of standard genotypes (STA; blue bars), families (FAM; red bars), and parents (PAR; grey bars) in control (CP) and defoliated plants (DP) across two growing seasons. Different lowercase letters indicate a significant difference between three groups (STA, FAM, and PAR) separately for each treatment (CP and DP); different uppercase letters indicate a significant difference between the two treatments separately for each group at $p < 0.05$ according to Tuckey's test. BMS—stem biomass, SPS—number of spikelets per spike, GNS—grain number per spike, GWS—grain weight per spike, TGW—1000-grain weight, HI—harvest index, GFR—grain-filling rate, SRC—stem reserve contribution to GWS.

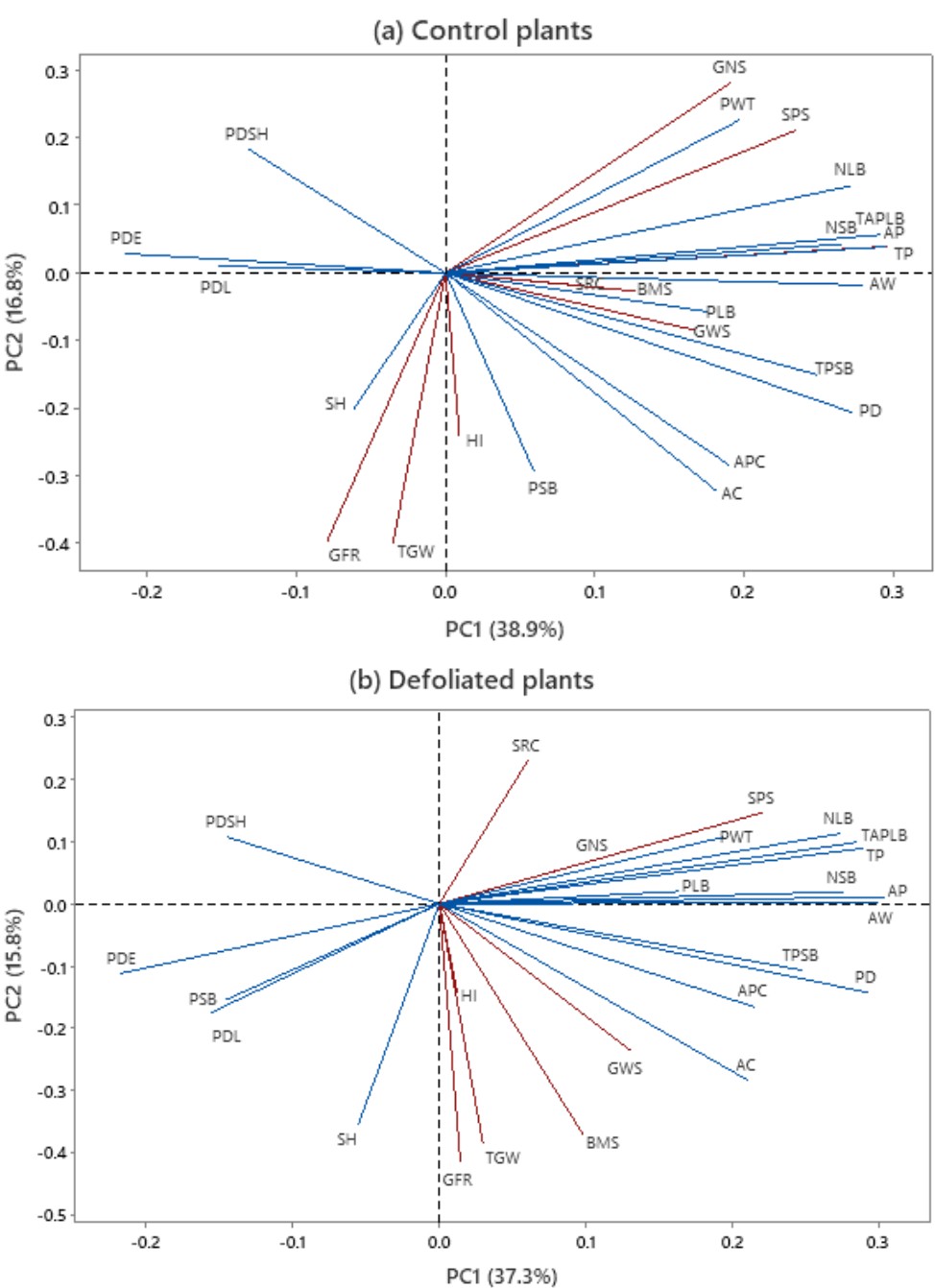

**Figure 3.** Genotype-by-trait biplot with interrelationship among traits in control (CP) and defoliated plants (DP) based on mean values across 20 genotypes and 2 growing seasons. Trait acronyms: SH—stem height, PDL—peduncle length, PDE—peduncle extrusion, PDS—peduncle share, BMS—main stem dry biomass, SPS—spikelets per spike, GNS—grain number per spike, HI—harvest index, TGW—1000-grain weight, SRC—stem reserve contribution, GRF—grain-filling rate, GWS—grain weight per spike, NSB—number of small vascular bundles, NSB—number of small vascular bundles, PD—peduncle diameter, PWT—peduncle wall thickness, AP—photosynthetically inactive parenchyma, AC—total area of chlorenchyma, PSB—average phloem area in one small vascular bundle, PLB—average phloem area in one large vascular bundle, TP—total area of phloem per stem transverse section, APC—area of pith cavity, and AW—area of peduncle wall.

## 4. Discussion

Despite the contrasting experimental weather conditions characterized by very high precipitation during the grain-filling stage in the second growing season (233 mm in May and 86 mm in June), across all cultivars, only the harvest index differed significantly in both treatments between the two growing seasons (Table 4). In addition, the stem reserve contribution in CP was significantly higher in the second growing season, indicating that stem carbon remobilization to the grains was affected by more than just plant water supply. Interestingly, the 1000-grain weight and grain weight per spike did not differ over the two seasons, although a temporal variation in the wheat grain yield commonly occurs in Europe due to weather conditions [33,34].

Based on our results for the two growing seasons and considering all the genotypes tested, the stem reserve contribution to grain weight per spike ranged from 5.2% to 99.0% in control and from 27.8% to 99% in defoliated plants, which is in accordance with previous studies [17,35,36]. Our results clearly showed that defoliation significantly reduced grain weight per spike, and increased the stem reserve contribution in both seasons and in all groups of genotypes (Table 4; Figure 2). Terminal drought shortens the grain filling period and increases the remobilization of assimilates from the stem to grains [3,10], but its effectiveness is typically limited by the shortening of the linear phase of grain development [17]. This may be the reason why there was no close association between grain weight per spike and stem reserve contribution in DP. Furthermore, the increased stem reserve contribution in defoliated plants was strongly associated with stem biomass. Interestingly, across all genotypes, grain number per spike was significantly reduced by defoliation in both growing seasons, and in the FAM group of genotypes across two growing seasons, which differs from the findings of previous studies [7,37]. Although it was previously shown that water stress imposed at anthesis through defoliation or during the lag stage of grain filling can reduce grain number per spike [38,39], our results presented here imply that the grain set could also be reduced when wheat is exposed to drought stress at 10 DAA, and this might be caused by the abortion of grains in later-developing florets in the spikelets. In contrast, there were no significant differences in spikelets per spike between the treatments either across all genotypes in both seasons or between the three groups of genotypes.

The grain-filling rate and duration have considerable effects on grain yield. It seems that very high precipitation during the grain-filling stage mitigated the effect of defoliation to a certain extent, as the 1000-grain weight and the grain-filling rate across all the genotypes remained unaffected in the second season. Although there was no leaf transpiration in the defoliated plants, it seems that under the conditions of very high precipitation, plants took up considerable amounts of water from the soil through their roots due to transpiration from other green aerial plant parts. Yet, markedly contrasting experimental weather conditions led to differential responses of wheat genotypes to defoliation. Both the 1000-grain weight and the grain-filling rate were significantly reduced by defoliation only in the PAR group of genotypes across the two growing seasons (Figure 2), as previously shown [40,41], indicating that those traits can be improved through the breeding process. In addition, grain weight per spike was strongly positively associated with these traits only in DP. The calculated stress indices SSI and STI show that the breeding lines from the FAM group are already more tolerant to terminal water stress conditions than the standard varieties (STA) and that they can be improved through further selection to exceed the stress tolerance of their parents.

From the anatomical point of view, the peduncle could have an important role in grain filling and yield, as it consists of photosynthetic tissue in wheat capable of synthesizing organic compounds, parenchyma cells with the capacity to store reserve carbohydrates, and vascular bundles, which serve as the path for remobilizing assimilates to the developing grains. The majority of previous studies dealing with the anatomy of the wheat stem have focused on the thickness of the stem wall, in relation to lodging resistance [42]. Thus, the anatomical structure of the wheat stem and its role in wheat tolerance to drought and grain filling has received less attention. The samples were taken for the measurement of stem

height and peduncle morpho-anatomical traits at 10 DAA, when the wheat stem reaches its maximum length [43], and stem anatomy is assumed to be complete [44]. Therefore, we did not compare peduncle morpho-anatomical traits in CP and DP. Our results indicated that FAM and PAR genotypes differed significantly in comparison with the two STA genotypes in some peduncle morpho-anatomical traits, but no significant differences were found between FAM and PAR genotypes overall, and variation in the anatomical traits of the two groups of genotypes was similar. Considering the associations between the traits, peduncle-length-related traits did not determine any of the productive ear traits or the stem reserve contribution and grain-filling rate. On the other hand, anatomical traits such as the area of chlorenchyma, peduncle diameter, and wall thickness, as well as the traits related to the phloem area, were positively associated with grain weight per spike. Apart from storage capacity, phloem transport, which depends on the sink size (grain weight/spike and grain number per spike), is also important for filling grains under drought conditions. A previous study on wheat stem anatomy indicated that the capacity for the accumulation and remobilization of dry matter from tissues in the wheat stem into grains may depend on stem wall thickness, the total phloem area, and the number of large vascular bundles, which were positively correlated with the compensatory effect to yield loss under mild and moderate water stresses [23].

**5. Conclusions**

In conclusion, the examination of wheat traits can provide significant information for breeding programs targeting wheat tolerance to drought stress. Besides the increased grain-filling rate and high stem biomass, the structure of the peduncle and the development of different tissues within it should be given more attention.

**Supplementary Materials:** The following supporting information can be downloaded at: https://www.mdpi.com/article/10.3390/agriculture13030673/s1, Table S1: Grain yield per spike, yield components, and stress susceptibility and stress tolerance index in control (CP) and defoliated plants (DP) of wheat genotypes across two growing seasons.

**Author Contributions:** Conceptualization, D.D., V.K., D.R. and J.S.; methodology, V.K., D.R., J.S. and D.D.; formal analysis, J.S., V.K., and D.R.; investigation, J.S., V.K., D.R. and D.D.; resources, V.K.; writing—original draft preparation, V.K., J.S. and D.R.; writing—review and editing, V.K., J.S. and D.R.; visualization, V.K., D.R., J.S. and D.D.; supervision, V.K.; project administration, V.K. All authors have read and agreed to the published version of the manuscript.

**Funding:** This research was funded by the Ministry of Science, Technological Development, and Innovation of the Republic of Serbia, grant number 451-03-47/2023-01/200116 and 451-03-47/2023-01/200040.

**Institutional Review Board Statement:** Not applicable.

**Data Availability Statement:** Not applicable.

**Acknowledgments:** We thank Radenko Radošević for technical assistance and microscope measurement and Steve Quarrie for kindly checking the text and correcting the English grammar and style.

**Conflicts of Interest:** The authors declare no conflict of interest.

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
