# Peer review of "Contribution of Agro-Physiological and Morpho-Anatomical Traits to Grain Yield of Wheat Genotypes under Post-Anthesis Stress Induced by Defoliation"

_agriculture, doi:10.3390/agriculture13030673_

Round 1

Reviewer 1 Report (Previous Reviewer 3)

The authors did good job with the manuscript correction and it can be published. 

Author Response

To the reviewer 1

We would like to thank to you for reading revised version of our paper and for suggestion that manuscript could be published.

Best regards,

Authors

Reviewer 2 Report (Previous Reviewer 4)

Dear authors,

You have done well. 

Although, the work is purposeful and distinctive, it done under uncontrolled conditions. In my view, to obtain reliable and consistent results, the experiments had been to conduct in the greenhouse experiments to control environmental conditions, especially in areas where weather conditions may differ significantly in two consecutive seasons. 

Best regards,

Author Response

We would like to thank to you and to reviewers for reading revised version of our paper and for additional comments and suggestions which improved our text. We agree that experiment should be done control environmental conditions, and we will try to do it in the future.

Thank you for your comment #7, but unfortunately we do not understand how to divide Figure 2 into two parts. This figure consists of 8 diagrams, and can you please explain us according to which criteria they should be divided into two parts?

Best regards,

Authors

This manuscript is a resubmission of an earlier submission. The following is a list of the peer review reports and author responses from that submission.

Round 1

Reviewer 1 Report

The authors of this article scored 20 wheat genotypes and 10 parental lines for their agro-physiological and peduncle morpho-anatomical traits to assess  the ability of these wheat genotypes to remobilize stem dry matter under water-stressed conditions simulated by defoliation 10 days after anthesis, and to estimate its contribution to grain weight. Their results show that the grain filling rate and 1000-grain wight as traits desirable for drought tolerance breeding.
  The materials and methods is apt and the results are sufficiently summarized and discussed. There are a couple of grammatical errors otherwise I do not have any other comment.

Author Response

Reviewer 1

Comments and Suggestions for Authors

The authors of this article scored 20 wheat genotypes and 10 parental lines for their agro-physiological and peduncle morpho-anatomical traits to assess  the ability of these wheat genotypes to remobilize stem dry matter under water-stressed conditions simulated by defoliation 10 days after anthesis, and to estimate its contribution to grain weight. Their results show that the grain filling rate and 1000-grain wight as traits desirable for drought tolerance breeding.

The materials and methods is apt and the results are sufficiently summarized and discussed. There are a couple of grammatical errors otherwise I do not have any other comment.

English is checked and improved. Thanks for your comments.

Reviewer 2 Report

Dear Authors,

I had the opportunity to read and review the manuscript entitled „Contribution of agro-physiological and morpho-anatomical traits to grain yield of wheat genotypes and their progenies under post-anthesis stress induced by defoliation” (ID agriculture-2167693).

This manuscript aims to assess the ability of 20 contrasting common wheat genotypes (two cultivars, eight F6:7 families (FAM) and ten parent genotypes (PAR) used to make families) to remobilize stem dry matter under water-stressed conditions simulated by defoliation 10 days after anthesis, and estimate its contribution to grain weight.

The manuscript is interesting and fits within the scope of Agriculture. The relevant aspects of the topic are present.

My review below suggests some improvements.

Title: the title matches the content.

Abstract: the abstract is clear and reasonable.

Keywords: keywords correspond to the aims and objectives of the manuscript.

1. Introduction: This section provides adequate insight into the research issues.

2. Materials and methods: experimental design and sampling are clearly described. However, the chapter should indicate some literature background for the method selection at the point of statistical analyses. As an example, I am not sure whether the t-test should be used to determine whether there are significant differences among the genotype groups. How do you control the significance level (the probability of Type I error) if you use the t-test instead of the ANOVA?

3. Results: The analysis of interactions between genotypes and treatments is missing from the study.

A color legend should be added to Figure 2. I found the explanation in the note below the figure, but I think the legend is more appropriate here.

Please check the variances in Figure 3: the variance explained by the first principal component (PC1) should be higher than the variance explained by the second principal component (PC2).

Some parameters are not clear to which group they belong: for instance, in CP, GNS, PWT SPS have close loadings in principal components 1 and 2. Therefore, it is not clear whether these traits belong to PC1 or PC2. To clarify the content of trait groups, I suggest using the Varimax rotation.

After clarifying the meaning of the first two principal components, it would be interesting to visualize the 20 contrasting common wheat genotypes in the coordinate system of the first two PCs.

4. Discussion: This section is quite detailed and discusses well results of the study compared to the other studies.

The article requires a major revision based on its revealed shortcomings (justification of the choice of methods, examination of interactions, verification and interpretation of PCA results).

Reviewer 3 Report

The paper is very original and based on solid two years of data with relevant methodology, arrives to significant results and conclusions. The study certainly represents and important contribution to the knowledge and worth publishing. However, the authors did not do good justice to the results and the paper has tremendous potential for improvement. The main pints are below.

1.       Line 20: “…whilst in both treatments grain filling rate was determination by stem, height.” Not clear what it means.

2.       Short information about wheat area, production and drought occurrence in Serbia would be very useful.

3.       Lines 75-76. “standard” is used two time in one sentence.

4.       Table 1. Parentage can be added for all material.

5.       Two important adaptation traits are plant height and number of days to heading. If the material varies widely for these two traits – the germplasm comparison may not be relevant. The authors may provide data on these traits in Table 3 or at least mention the variation.

6.       Sections 3.1 and 3.3. The comparison between STA, FAM and PAR groups is not well justified. Why to compare them? To document breeding progress – too few lines. If FAM groups was subjected to specific drought selection process – perhaps justified but need to be explained. The only justifiable grouping of this material is Serbian versus foreign or modern versus old. The FAM lines can not be compared to parents for inheritance evaluation since such comparison requires a relatively large number of lines and may have different experimental design.

7.       Section 3.4. The authors presented biplot based on average data for two seasons despite the fact that the seasons were very different. It is worth to conduct the biplot separate for each year and see if some interesting interaction are missed by using mean values.

8.       Lines 255-256. “grain filling” is two times in one sentence, “draught” means drought.

9.       Discussion can be improved by removing repetition of results and references to tables and figures. Broader view of the study outcomes and their implication for physiology and practical breeding would be very useful.

10.   The authors need to present additional table with superior genotypes considering grain yield, its components and response to the treatment.

11.   There is excessive number of abbreviations. Normally traits with three words can be abbreviated. One or two words require no abbreviations. Certainly STA, FAM and PAR can be written in words. The treatments require no abbreviation.  

12.   Overall, the main challenge of the paper is comparison of three groups because their intrinsic differences are not sufficiently described. Instead, it may be better to present the results with the view of identification of superior drought tolerant genotypes.

13.   The title may be changed to: “Contribution of agro-physiological and morpho-anatomical traits to grain yield of wheat genotypes under post-anthesis stress induced by defoliation”. The paper does not provide any linkage between the parents and progenies. Hence, they can be all treated as a set of independent genotypes compared to checks.

Reviewer 4 Report

Thank you for your invitation to review the manuscript entitled "Contribution of agro-physiological and morpho-anatomical traits to grain yield of wheat genotypes and their progenies under post-anthesis stress induced by defoliation".

Although the topic is interesting. The results are not sufficient.   Anyway, the authors worked on the exposure of wheat plants to drought stress and defoliated the drought-stressed plants. They studied some anatomical and morphological attributes. The manuscript has no novelty. Theoretically, when the plants are defoliated, they will change their morphological, physiological, and anatomical characteristics to survive. The anatomical part is expected and acceptable. In the physiological traits (Table 4), there are no marked differences in yield components between the control and the defoliated and drought-stressed plants. These results are unexpected theoretically and unacceptable in my view. Therefore, the research needs more physiological analysis to prove how the defoliated and drought-stressed wheat plants (herbaceous stem) perform and produce yield like the normal plants. In my view, the results may need more checking.
